# Idiopathic hyperprolactinemia-associated hypogonadism in men presenting with normal testosterone levels

**Xiaozhi Cheng**[1,2]*, **Yunbei Xiao**[1], **Yulong Deng**[2], **Qiwen Chen**[2], **Xiaoyan Wen**[3], **Er Zhou**[2]*, **Huiliang Zhou**[1]*

**1** Department of Andrology and Sexual Medicine, The First Affiliated Hospital of Fujian Medical University, Fuzhou, Fujian, China, **2** Department of Neurosurgery, People's Hospital of Gaozhou, Maoming, Guangdong, China, **3** Department of Finance and Asset Management, People's Hospital of Gaozhou, Maoming, Guangdong, China

\* cxz@fjmu.edu.cn (XZC); zhlpaper@fjmu.edu.cn (HLZ); 1743619419@qq.com (EZ)

## Abstract

### Purpose

Hypogonadism, presenting with low libido, erectile dysfunction, and gynecomastia, frequently occurs in men with hyperprolactinemia, typically characterized by elevated serum prolactin, suppressed gonadotropins, and low testosterone. However, we identified a rare subset of patients who presented with normal testosterone levels, but whose clinical profiles were poorly defined.

### Methods

A retrospective analysis of medical records was conducted on 23 men diagnosed with idiopathic hyperprolactinemia and normal testosterone levels between 01/07/2019 and 01/12/2024. Baseline clinical characteristics, hormone levels, pituitary gland dimensions (as measured by MRI-derived pituitary height), and responses to medical treatments (bromocriptine or cabergoline) were evaluated.

### Results

The mean age at diagnosis was 30.91 ± 7.66 years (range: 20–50). Mean serum prolactin at presentation was 38.22 ± 30.68 ng/mL (range: 20.00–170.93), and mean pituitary height was 6.40 ± 0.97 mm. Primary complaints at diagnosis included low libido, gynecomastia, impotence, and erectile dysfunction. Eighteen patients received bromocriptine, whereas five patients received cabergoline. After an average treatment duration of 7.83 ± 6.97 months (range: 1–29 months), prolactin levels normalized in all patients, pituitary height was significantly reduced to 4.39 ± 1.37 mm, and most patients reported notable clinical improvements, particularly in sexual function.

**Data availability statement:** All relevant data are within the manuscript and its Supporting Information files.

**Funding:** This study was supported by the Medical Scientific Research Foundation of Guangdong Province, China (No. B2024265, No. B2025362).

**Competing interests:** The authors have declared that no competing interests exist.

## Conclusion

Idiopathic hyperprolactinemia-associated hypogonadism can occasionally present with normal testosterone levels, emphasizing the importance of clinical vigilance beyond standard hormonal assessments. Pituitary height reduction may serve as an important diagnostic marker and indicator of treatment effectiveness. Medical treatment, including bromocriptine or cabergoline, with a possible preference for cabergoline based on clinical considerations, can effectively normalize hormone levels and significantly improve clinical symptoms.

## Introduction

Hyperprolactinemia (HPRL) was proposed by Corona *et al.* as a new clinical syndrome associated with sexual dysfunction [1]. All types of hyperprolactinemia (idiopathic, tumoral or drug-induced) can impede various facets of male sexual behavior. Therefore, although HPRL is a rather rare condition, it can lead to male sexual dysfunction and should not be overlooked due to its potential reversibility [2].

Hypogonadism, characterized by decreased libido, erectile dysfunction (ED), gynecomastia, is a primary symptom of HPRL in men, and typically manifests with elevated serum prolactin (PRL) levels, suppressed gonadotropin levels, and low testosterone levels [3]. In a clinical trial, the correction of prolactin levels in male patients with severe hyperprolactinemia using medications such as cabergoline (CAB) or bromocriptine (BRC) improved hypogonadism-related symptoms [4]. The findings from this clinical trial, along with other studies, have provided evidence that elevated prolactin levels or HPRL are involved in the pathogenesis of sexual dysfunction in men [5,6].

It is widely recognized that HPRL-induced sexual dysfunction involves a decrease in testosterone secretion, which can affect androgen-dependent pathways [7,8]. Testosterone is converted by 5α-reductase into the most potent androgen dihydrotestosterone (DHT), which binds to androgen receptor (AR) to exert its promotive effects on penile erection and libido [9]. However, some men with sexual dysfunction and HPRL exhibit normal serum testosterone levels [3,10], suggesting mechanisms unrelated to testosterone/androgens can also contribute to the development of sexual dysfunction.

Idiopathic HPRL is characterized by mild to moderate elevated prolactin levels, normal pituitary MRI imaging without a visible adenoma, and excludes drug-induced HPRL, macroprolactinemia, chronic renal failure, and primary hypothyroidism [11]. Given the rarity of idiopathic HPRL in males, we present our experience with the diagnosis and treatment of 23 male patients with HPRL alongside normal testosterone levels.

## Patients and methods

### Patients

We retrospectively screened data from 234 men with HPRL who were admitted to People's Hospital of Gaozhou (Gaozhou, Guangdong, China) between 01/07/2019

and 01/12/2024. Ultimately, 23 patients were retrospectively enrolled in this study with idiopathic HPRL and normal testosterone levels. Their medical records were reviewed for clinical characteristics, signs and symptoms, International Index of Erectile Function (IIEF)-15 score, laboratory tests, MRI scans, treatment approach, duration of treatment and response to treatment. Informed consent was obtained from all patients prior to treatment, and all patients agreed to the use of their clinical data. This retrospective study was conducted in accordance with the Declaration of Helsinki and data collection was anonymous, approved by the Ethics Committee of People's Hospital of Gaozhou on 25/07/2024 (approval No. GYLLPJ-2024080), which waived the requirement for further informed consent.

All patients were treated with bromocriptine, except for five patients who were intolerant to bromocriptine switched to cabergoline. The initial oral dosage of bromocriptine was 2.5 mg twice daily, with subsequent adjustments based on hormonal reactions. Serum total testosterone levels were monitored alongside PRL levels every one to three months in the initial year post-diagnosis, with a reduced frequency thereafter to semiannual or as clinically indicated. Normalization of PRL levels determined the efficacy of PRL secretion control.

### Laboratory assays

PRL and total testosterone levels were measured using standard procedures and commercial kits obtained from Roche Diagnostics (Basel, Switzerland). In our laboratory, the reference range for PRL levels in men is 4.04–15.2 ng/mL, with levels exceeding 200 ng/mL were determined following appropriate serum dilutions. For total testosterone, the reference range for men is 2.49–8.36 ng/mL.

Baseline serum morning PRL levels and total testosterone levels were measured 2–3 hours after waking from sleep and repeated at least twice, drug-induced hyperprolactinemia was excluded, and macroprolactinemia was eliminated if the clinical scenario was appropriate.

### Pituitary imaging

All patients underwent sellar 3.0-Tesla MRI(DiscoveryMR750, GE) before initiation of dopamine agonist treatment and additional MRI scans performed during treatment based on clinical requirements.The pituitary height was measured as the craniocaudal distance from the inferior border of the gland to the insertion point of the pituitary stalk in the mid-sagittal plane, defined as the plane where the anterior lobe, posterior lobe, and pituitary stalk were all clearly visible.

### IIEF-15 score

The IIEF-15 is a self-administered questionnaire comprising 15 items that evaluate five key domains of male sexual function: erectile function, orgasmic function, sexual desire, intercourse satisfaction, and overall satisfaction in the past 4 weeks. Erectile function is evaluated through six questions (questions 1–5 and 15), with a maximum score of 30 points. Men scoring below 26 are categorized as having mild (22–25), mild to moderate (17–21), moderate (11–16), or severe (10 or less) erectile dysfunction. Intercourse satisfaction is determined by three questions (questions 6–8), while the other domains are evaluated using two questions each (orgasmic function: questions 9 and 10, sexual desire: questions 11 and 12, overall satisfaction: questions 13 and 14). The maximum scores are 15 for intercourse satisfaction and 10 for orgasmic function, sexual desire, and overall satisfaction. Conversely, the minimum scores were 0 for intercourse satisfaction and orgasmic function, 1 for erectile function, and 2 for sexual desire and overall satisfaction.

### Statistical analysis

Statistical analysis was performed using GraphPad Prism 8 (GraphPad Software, USA). Continuous variables were expressed as the mean ± standard deviation and compared using the Wilcoxon signed rank test and Mann–Whitney U test. Spearman rho correlation coefficient and linear regression analysis were employed to examine the associations between prolactin levels and pituitary height. $P < 0.05$ was considered statistically significant.

## Results

### Patients' characteristics at presentation

The study cohort included 23 men with idiopathic hyperprolactinemia and normal testosterone levels identified and followed between July 2019 and December 2024. This subset represents approximately 10% of all men with hyperprolactinemia evaluated during the study period, with the majority (~90%) exhibiting the typical finding of reduced testosterone levels. Mean age at diagnosis was 30.91 ± 7.67 (range, 20–50) years (Table 1). Mean serum PRL level at presentation was 38.22 ± 30.68 ng/mL (range, 20.00–170.93). Eighteen of the men presented with PRL levels less than or equal to three times the upper limit of normal (≤ 45.6 ng/mL). Mean baseline testosterone was 5.16 ± 1.86 ng/mL (normal, 2.49–8.36 ng/mL). Initial complaints leading to diagnosis included low libido, weakness, gynecomastia and ED. Mean IIEF-15 score was 52.96 ± 11.19(range, 28–69). IIEF-15 score was low in most patients, but 9 men presented with normal sexual function (IIEF-15 ≥ 60). All patients in the cohort had intact pituitary function. Thyroid function tests were normal in all patients. Initial MRI scans revealed no structural abnormalities in the pituitary gland of any patient, mean baseline pituitary height was 6.40 ± 0,97 mm (range, 5.1–8.6). Twelve men underwent repeated imaging. The baseline clinical characteristics of the men included in the cohort are presented in Table 1.

### Medical treatment

Medical treatment with dopamine agonists was given to all affected men following diagnosis. All patients were treated with bromocriptine, except for five patients who were intolerant to bromocriptine switched to cabergoline (0.5 mg/week). Starting dose of bromocriptine was usually 2.5 mg/day, and maximal dose was 7.5 mg/day. Treatment with dopamine agonists achieved PRL normalization in all patients (Table 1). Mean baseline PRL decreased from 38.22 ± 30.68 to 8.88 ± 3.04 ng/mL (p < 0.05, Fig 1A). Mean testosterone increased from a baseline of 5.52 ± 1.86 to 5.74 ± 2.07 ng/mL with no difference between before and after treatment(p > 0.05, Fig 1B). Meanwhile the mean IIEF-15 score was significantly increased by treatment (52.96 ± 11.19 versus 65.48 ± 8.28, $P < 0.05$, Fig 1C). Improvement in sexual function(the number of patients with IIEF-15 scores <60 increased to ≥60) was observed in 64.28% (9/14) of patients who received medication, including improvements in all domains of erectile function, orgasm function, sexual desire, sexual intercourse satisfaction, and overall satisfaction(Fig 1D). Clinically, most patients reported improvement of their low libido complaints following medical treatment, including seven patients with ED, and one patient experienced resolution of gynecomastia. Two patients with ED show clinical improvement(Table 1, patient 7 and patient 15).

### Pituitary height and prolactin normalization

Patients underwent magnetic resonance imaging (MRI) prior to dopamine agonist therapy, and pituitary height was subsequently measured. Twelve participants underwent repeated MRI assessments due to an extended treatment regimen. Pituitary height was documented upon normalization of prolactin levels (Fig 2A–D).

MRI scans of all participants during the treatment period revealed no pathological conditions, such as pituitary tumors or empty sella syndrome. The mean baseline pituitary height was 6.40 ± 0.97 mm, which significantly decreased to 4.39 ± 1.37 mm post-treatment (P < 0.05, Fig 3A). Correlation analysis demonstrated a positive correlation between prolactin levels and pituitary height (r = 0.78, P < 0.0001, Fig 3B).

### Duration of treatment

The duration for prolactin normalization varied from 1 to 29 months, with a mean of 7.83 ± 6.97 months. Of the patients, 18 patients received bromocriptine therapy, while 5 switched to cabergoline. The mean treatment duration for cabergoline (2.40 ± 1.14 months) was significantly shorter than that for bromocriptine (9.33 ± 7.18 months) (P < 0.05, Fig 4A). Additionally, cabergoline demonstrated a greater capacity to lower prolactin levels and enhance the IIEF-15 score, although these differences did not reach statistical significance (Fig 4B-C).

**Table 1. Baseline characteristics and response to medical treatment of each patient in the cohort.**

| No | Age (years) | Pre-treatment | | | | | Max BRC dose mg/day | Post-treatment | | | | | Duration of treatment (months) |
|---|---|---|---|---|---|---|---|---|---|---|---|---|---|
| | | PRL (4.04–15.2) ng/mL | T (2.49–8.36) ng/mL | Initial Complaints | IIEF-15 score | Pituitary Height(mm) | | PRL (4.04–15.2) ng/mL | T (2.49–8.36) ng/mL | Complaints improved | IIEF-15 score | Pituitary Height(mm) | |
| 1 | 28 | 40.00 | 6.66 | Low libido | 60 | 5.9 | 7.5 | 9.32 | 4.25 | Libido | 70 | 5.0 | 19 |
| 2 | 20 | 170.93 | 8.43 | Low libido, ED, Gyne-comastia | 28 | 8.6 | 10 | 5.76 | 5.74 | Gyneco-mastia | 58 | 5.1 | 29 |
| 3 | 29 | 52.81 | 5.14 | Low libido | 53 | 6.1 | 7.5 | 10.33 | 9.88 | Libido | 69 | 4.5 | 14 |
| 4 | 31 | 44.31 | 8.00 | Low libido | 63 | 6.6 | 7.5 | 13.74 | 4.13 | Libido | 70 | 4.7 | 16 |
| 5 | 30 | 29.45 | 6.98 | Low libido | 65 | 5.5 | CAB 0.5 mg/week | 11.07 | 8.12 | Libido | 73 | N/A | 2 |
| 6 | 30 | 42.69 | 6.40 | Low libido | 53 | 7.4 | 7.5 | 6.37 | 7.65 | Libido | 63 | N/A | 6 |
| 7 | 24 | 47.20 | 4.53 | ED | 42 | 6.3 | 7.5 | 4.53 | 6.11 | Libido | 53 | 4.9 | 10 |
| 8 | 50 | 22.49 | 3.49 | Low libido | 69 | 5.9 | 5 | 8.2 | 2.95 | Libido | 73 | N/A | 3 |
| 9 | 24 | 20.30 | 6.66 | Low libido, Weakness | 51 | 6.2 | 5 | 9.83 | 3.08 | Libido, Weakness | 66 | N/A | 3 |
| 10 | 30 | 23.93 | 3.48 | Low libido | 60 | 6.2 | 5 | 6.15 | 7.57 | Libido | 72 | N/A | 4 |
| 11 | 30 | 49.86 | 4.86 | Low libido | 61 | 7.5 | CAB 0.5 mg/week | 8.46 | 8.14 | Libido | 72 | N/A | 1 |
| 12 | 30 | 26.75 | 5.18 | Low libido | 67 | 5.1 | 5 | 9.58 | 6.32 | Libido | 72 | N/A | 1 |
| 13 | 25 | 23.70 | 3.67 | ED | 43 | 6.7 | 7.5 | 14.4 | 8.23 | Libido, ED | 58 | 4.3 | 7 |
| 14 | 30 | 20.00 | 2.66 | ED | 35 | 5.2 | 2.5 | 10.71 | 3.63 | Libido, ED | 52 | N/A | 2 |
| 15 | 30 | 25.40 | 5.78 | ED | 37 | 5.6 | CAB 0.5 mg/week | 7.52 | 8.3 | Libido | 41 | N/A | 2 |
| 16 | 35 | 29.60 | 3.00 | Low libido | 67 | 5.2 | 5 | 4.06 | 7.45 | Libido | 72 | 4.3 | 4 |
| 17 | 40 | 27.40 | 4.93 | ED | 51 | 5.9 | CAB 0.5 mg/week | 9.66 | 4.33 | Libido, ED | 72 | N/A | 4 |
| 18 | 31 | 36.80 | 2.51 | ED | 48 | 7.1 | 7.5 | 11.22 | 4.51 | ED | 61 | 4.5 | 6 |
| 19 | 39 | 20.50 | 4.07 | ED | 54 | 5.5 | 7.5 | 9.87 | 5.43 | Libido, ED | 67 | 5.1 | 12 |
| 20 | 26 | 26.80 | 6.11 | Low libido | 64 | 6.3 | 7.5 | 14.41 | 2.73 | Libido | 72 | 5.5 | 9 |
| 21 | 50 | 28.50 | 2.71 | ED | 52 | 7.6 | 7.5 | 3.84 | 4.92 | Libido, ED | 67 | 4.6 | 15 |
| 22 | 28 | 43.70 | 8.73 | ED | 42 | 6.5 | CAB 0.5 mg/week | 7.35 | 4.65 | ED | 68 | N/A | 3 |
| 23 | 21 | 26.00 | 4.65 | ED | 53 | 8.3 | 7.5 | 6.93 | 3.82 | Libido, ED | 65 | 5.2 | 8 |
| Mean | 30.91 | 38.22 | 5.158 | – | 52.96 | 6.400 | – | 8.840 | 5.737 | – | 65.48 | 4.392 | 7.826 |
| SD | 7.663 | 30.68 | 1.857 | – | 11.19 | 0.9653 | – | 3.039 | 2.067 | – | 8.279 | 1.369 | 6.972 |

BRC bromocriptine, CAB cabergoline, ED erectile dysfunction, IIEF-15 International Index of Erectile Function-15, N/A not available, PRL prolactin, SD standard deviation, T testosterone

## Discussion

This study reports the treatment outcomes of 23 patients with idiopathic hyperprolactinemia and normal testosterone levels. Three key findings emerged from our analysis. (1) The primary symptoms were hypogonadism (libido loss, erectile dysfunction, gynecomastia), with mildly elevated prolactin levels that responded well to dopamine agonist therapy; the hypogonadism symptoms improved concurrently with the reduction in prolactin. (2) Serum prolactin levels positively

(A)

(B)

(C)

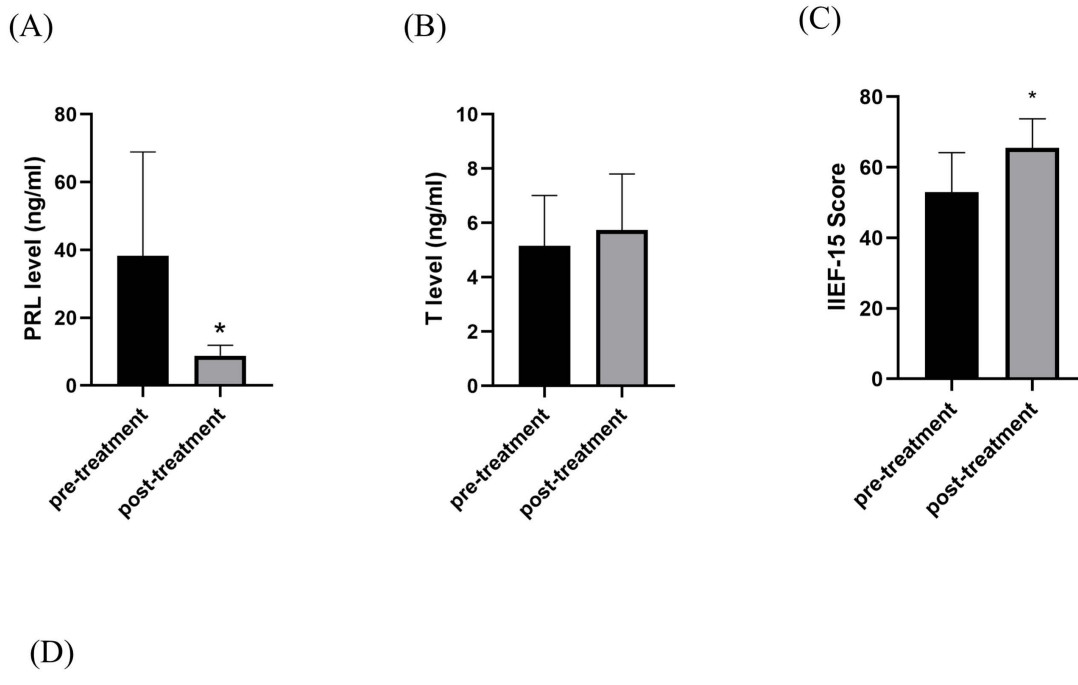

(D)

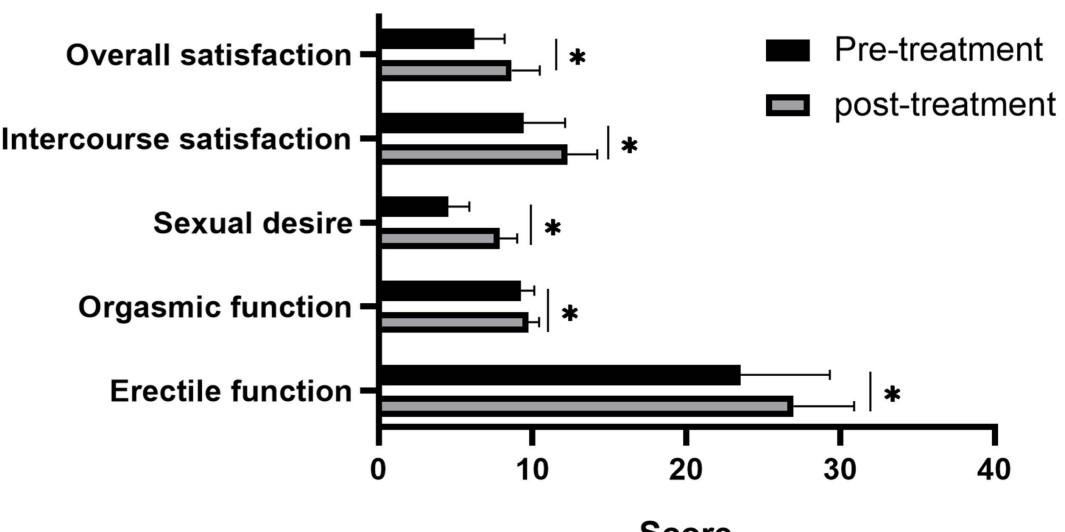

**Fig 1. Changes in prolactin levels, testosterone levels, IIEF-5 scores and sexual functioning after treatment with dopamine agonists.** Among 23 patients with idiopathic hyperprolactinemia and normal testosterone levels, Eighteen received bromocriptine treatment, whereas five switched to cabergoline. **(A)** Prolactin levels before and after treatment. **(B)** Testosterone levels before and after treatment. **(C)** IIEF-15 scores before and after treatment. **(D)** Five key domains scores of male sexual function before and after treatment. IIEF-15, International Index of Erectile Function-15; PRL, prolactin; T, testosterone. *$P < 0.05$.

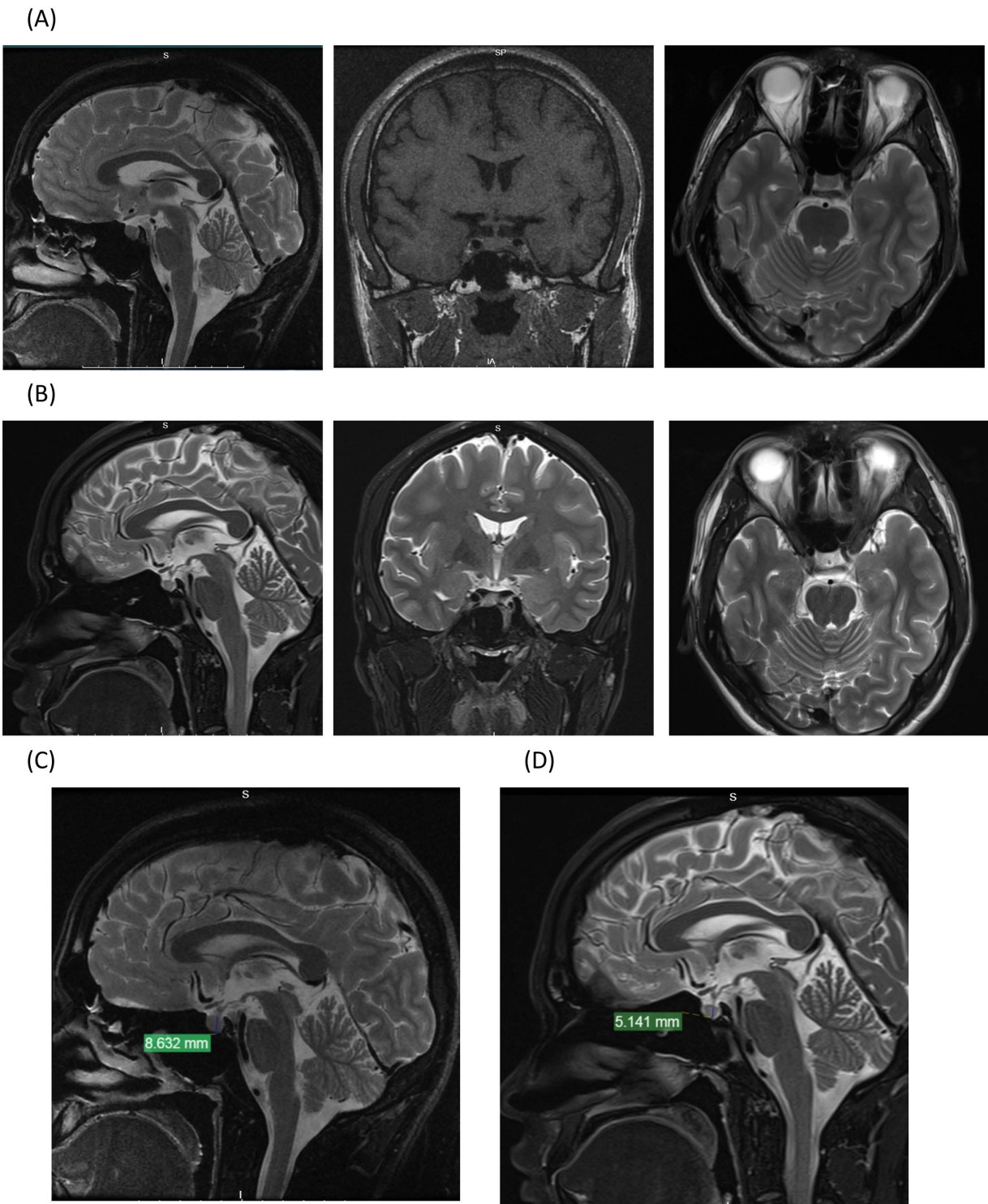

**Fig 2. Sellar magnetic resonance imaging (MRI) scans and measurement of pituitary height before and after treatment.** All Patients underwent MRI prior to dopamine agonist therapy, and pituitary height was subsequently measured. Twelve participants underwent repeated MRI assessments due to an extended treatment regimen. Sagittal, coronal, and axial views of MRI scan of the sellar region before (A) and after treatment **(B)**. Sagittal pituitary height on MRI was 8.632 mm before treatment **(C)**, compared to 5.144 mm after treatment **(D)**.

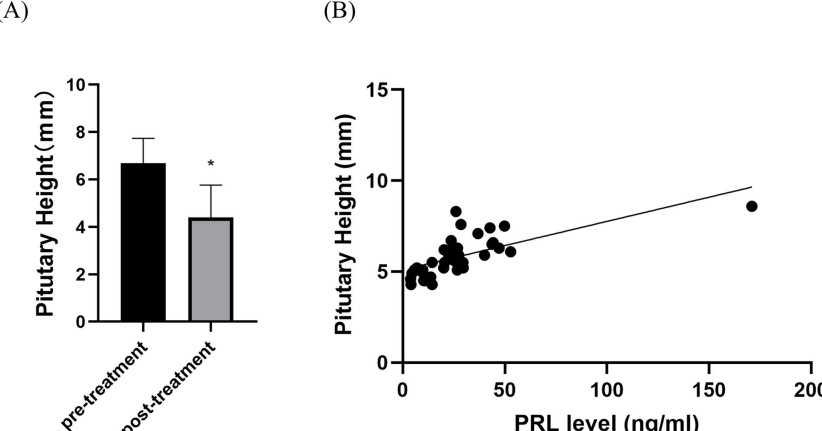

(A)                          (B)

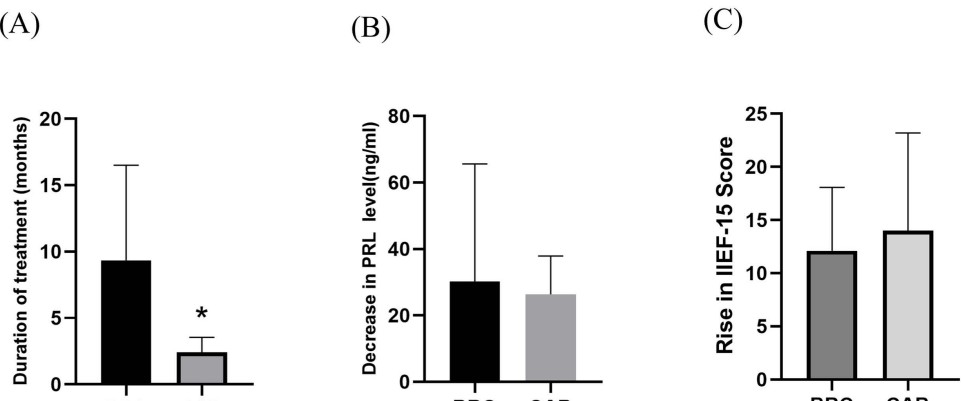

**Fig 3. Changes in pituitary height after treatment and its correlation with prolactin. (A)** Pituitary height before and after treatment. **(B)** Correlation between prolactin levels and pituitary height in patients with idiopathic hyperprolactinemia, Correlation analysis revealed a positive linear correlation between PRL levels and prolactin levels and pituitary height (r = 0.7823, P < 0.0001). PRL, prolactin. *P < 0.05.

(A)                    (B)                    (C)

**Fig 4. Comparative analysis of cabergoline and bromocriptine efficacy in patients with idiopathic hyperprolactinemia and normal testosterone levels. (A)** Cabergoline normalized prolactin levels more rapidly than bromocriptine. **(B)** Although not statistically significant, cabergoline demonstrated a more pronounced reduction in prolactin levels compared to bromocriptine. **(C)** Cabergoline provided greater improvement in IIEF-15 scores relative to bromocriptine, though this difference also lacked statistical significance. BRC, bromocriptine. CAB, cabergoline. *P < 0.05.

correlated with pituitary height, which also decreased after prolactin normalization. (3) Cabergoline achieved prolactin normalization more rapidly than bromocriptine in managing idiopathic hyperprolactinemia. These findings may inform new diagnostic and therapeutic strategies for hypogonadism patients with idiopathic hyperprolactinemia and normal testosterone levels.

Prolactin, secreted by anterior pituitary lactotrophs, plays a multifaceted role in mammalian physiology far beyond its classical association with lactation.[12] Beyond its well-known function in the regulation of lactation, prolactin acts as a pleiotropic factor, playing diverse roles in human health and disease through its specific binding to the transmembrane prolactin receptor, which is expressed in a wide range of tissues, including the reproductive, immune, vascular, and central nervous systems. [12,13] In terms of reproductive health and disease in men, prolactin has been implicated in erectile function and sexual behavior. Elevated levels of prolactin, as predominately observed in conditions such as prolactinoma,

can disrupt these processes, resulting in symptoms such as ED and infertility. In line with previous studies, [6,12,14–16] our results indicated that individuals with idiopathic hyperprolactinemia frequently experience low libido, erectile dysfunction, weakness, and gynecomastia. Medication treatment significantly alleviated these symptoms. Additionally, our study showed that hypogonadism associated with idiopathic hyperprolactinemia can occur even in the absence of decreased testosterone levels.

The mechanisms underlying hyperprolactinemia-induced hypogonadism are complex and multifaceted, involving both the central nervous and peripheral systems. Elevated prolactin levels can disrupt the balance of neurotransmitters critical for sexual arousal and erectile function, such as dopamine, which is known to enhance libido and erectile responses. [16] Increased prolactin levels can lead to reduced dopaminergic activity, contributing to diminished sexual function. Additionally, hyperprolactinemia can affect the vascular and endothelial functions necessary for achieving and maintaining an erection. Elevated prolactin levels can lead to endothelial dysfunction, characterized by impaired NO production, which is essential for vascular relaxation and increased blood flow to the penis. [17,18] This dysfunction can result in inadequate penile perfusion, leading to difficulties in achieving an erection.

Hypogonadism in men with prolactinomas is not solely related to elevated prolactin levels, but also closely associated with testosterone deficiency. [19] Interestingly, our study cohort exhibited normal testosterone levels despite elevated prolactin levels in the absence of a detectable adenoma. Similar cases have been previously documented. Nevertheless, we observed an upward trend in testosterone levels following treatment with dopamine receptor agonists, although the change was not statistically significant (Fig 1B). This may indicate a relative insufficiency of testosterone despite being in the normal range. In other words, "normal" circulating testosterone in these patients did not guarantee normal androgen effect – a concept supported by their symptomatic improvement when prolactin was lowered and testosterone rose slightly. Normalizing prolactin levels appears to be necessary to relieve the inhibitory effect on testosterone production. In our cohort, baseline prolactin levels were mildly elevated, with a mean of $38.22 \pm 30.68$ ng/mL. This indicates that the pituitary-gonadal axis may remain intact in these patients, allowing them to respond well to dopamine agonist treatment. This is in contrast to patients with invasive prolactinomas, whose pituitary-gonadal axis may be permanently damaged, requiring testosterone replacement therapy even after normalization of prolactin levels.

Pituitary MRI represents the most effective imaging modality for the diagnosis of hypothalamic-pituitary disorders. [20] However, the routine use of this technique in the evaluation of hypogonadism remains debated, as pituitary structural abnormalities occur infrequently and MRI testing is costly. [21] Nonetheless, several studies have demonstrated the utility of pituitary MRI in the assessment of male patients with hypogonadism. [22,23] The existing literature has primarily focused on the relationship between pituitary structural abnormalities and hypogonadism, while overlooking the potential association between pituitary height and gonadal hormones in the context of a normal pituitary structure. Pituitary height is currently used primarily to evaluate growth and development in children. [24,25] In the present study cohort, we observed a positive correlation between prolactin levels and pituitary height in patients with idiopathic hyperprolactinemia-associated hypogonadism, such that higher prolactin levels were accompanied by increased pituitary height. This finding is consistent with the research of Argyropoulou and colleagues, [25] who reported that increased pituitary height in premature infants suggests a more immature hypothalamic-pituitary axis, as evidenced by enhanced secretory activity of the adenohypophysis. Importantly, we also found that pituitary height returned to the normal range of $4.39 \pm 1.37$ mm following pharmacological treatment. Interestingly, these values align with the findings of Berntsen and colleagues, [26] who reported a mean mid-sagittal pituitary height of $4.4 \pm 1.4$ mm in a cohort of 388 healthy men aged 50–66 years. We postulate that pituitary height might serve as a dynamic marker of prolactin secretion activity even in the absence of a tumor. This is a novel observation that warrants further investigation.

Dopamine agonists, commonly prescribed for the pharmacological management of hyperprolactinemia, appear to influence sexual functioning, particularly in men. [27] Cabergoline and bromocriptine have been shown to enhance erectile response in men with hyperprolactinemia. [28] Furthermore, cabergoline treatment has been found to improve sexual

drive and function, as well as positively impact the perception of the refractory period. [4] Our findings corroborate these observations. Based on the current evidence, cabergoline appears to be a highly effective treatment option for patients with hyperprolactinemia compared to bromocriptine; [29] however, our results did not show a statistically significant difference in efficacy between these agents. Cabergoline's longer half-life and better tolerability often make it the preferred first-line agent in clinical practice [29], but our data emphasize that when it comes to outcomes like prolactin normalization and symptom improvement, bromocriptine can be similarly effective in idiopathic cases. Thus, the choice of agent may be guided by side effect profile and patient preference rather than efficacy alone in this specific scenario.

Limitations: The present study has several limitations that warrant consideration. First, while the IIEF-15 questionnaire is a well-validated instrument for assessing sexual function, its utility is constrained by inherent subjectivity and reliance on patient self-report. Objective measures of sexual function were not utilized, which could introduce bias. Second, the retrospective design and modest sample size (only 23 patients) limit the strength and generalizability of our conclusions. A small sample increases the risk of Type II error (missing true effects) and may not capture the full spectrum of this condition. Third, our study was conducted at a single center, and all patients were Han Chinese men; thus, the findings may not be fully generalizable to other populations. Future prospective, multi-center studies are needed to confirm these findings in larger, more diverse cohorts. Such studies should also specifically evaluate pituitary height as a biomarker for idiopathic hyperprolactinemia-associated hypogonadism, to determine if this measure can reliably aid in diagnosis or tracking of treatment response across different settings..

## Conclusion

The present study demonstrates that a subset of male patients with idiopathic hyperprolactinemia-associated hypogonadism exhibit normal testosterone levels. Pituitary height can serve as a diagnostic marker and predictor of treatment efficacy. Dopaminergic therapy (cabergoline or bromocriptine) not only normalizes prolactin levels but also restores sexual function and pituitary morphology, underscoring its value even in normotestosteronemic patients, In our study, cabergoline normalized prolactin more rapidly, but overall outcomes were similar to bromocriptine, with no significant advantage of one over the other. Our findings challenge the conventional reliance on testosterone levels alone in the evaluation of male sexual dysfunction and advocate for a broader hormonal and radiological assessment.

## Supporting information

**S1 Data.  IIEF-15 Summary sheet.**
(XLSX)

## Author contributions

**Conceptualization:** Er Zhou, Huiliang Zhou.

**Data curation:** Xiaozhi Cheng, Yunbei Xiao, Yulong Deng, Qiwen Chen.

**Formal analysis:** Xiaozhi Cheng, Yunbei Xiao, Yulong Deng, Qiwen Chen, Xiaoyan Wen.

**Funding acquisition:** Xiaozhi Cheng.

**Methodology:** Xiaozhi Cheng, Yulong Deng, Qiwen Chen, Xiaoyan Wen.

**Project administration:** Xiaozhi Cheng, Huiliang Zhou.

**Software:** Xiaozhi Cheng, Xiaoyan Wen.

**Supervision:** Xiaozhi Cheng, Er Zhou, Huiliang Zhou.

**Writing – original draft:** Xiaozhi Cheng.

**Writing – review & editing:** Xiaozhi Cheng, Yunbei Xiao, Yulong Deng, Qiwen Chen, Er Zhou, Huiliang Zhou.

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
