## [Decision Letter · Decision Letter 0]

3 Sep 2025

PONE-D-25-39935Idiopathic hyperprolactinemia-associated hypogonadism in men presenting with normal testosterone levelsPLOS ONE

Dear Dr. Cheng,

Thank you for submitting your manuscript to PLOS ONE. After careful consideration, we feel that it has merit but does not fully meet PLOS ONE’s publication criteria as it currently stands. Therefore, we invite you to submit a revised version of the manuscript that addresses the points raised during the review process.

**ACADEMIC EDITOR: **

Your manuscript is of interest and well-structured. Please revise by clarifying the sentence at lines 169–170, moderating conclusions on cabergoline (as differences were not statistically significant), and improving grammar, formatting, and consistency (spelling, units, abbreviations). Expand on the limitations (retrospective design, small sample, use of IIEF-15) and suggest future directions. Ensure the ethics approval date is included, and provide higher resolution images if available.

We look forward to receiving your revised manuscript.

Kind regards,

Ghadeer Sabah Bustani, Ph.D

Academic Editor

PLOS ONE

 [This study was supported by the Medical Scientific Research Foundation of Guangdong Province, China (No. B2024265, No. B2025362).]. 

Additional Editor Comments (if provided):

Reviewer #1:

Reviewer #2:

Reviewer #3:

Reviewer #4:

Reviewers' comments:

Reviewer's Responses to Questions

**Comments to the Author**

1. Is the manuscript technically sound, and do the data support the conclusions?

Reviewer #1: Yes

Reviewer #2: Yes

Reviewer #3: Yes

Reviewer #4: Yes

2. Has the statistical analysis been performed appropriately and rigorously? 

Reviewer #1: Yes

Reviewer #2: I Don't Know

Reviewer #3: Yes

Reviewer #4: Yes

3. Have the authors made all data underlying the findings in their manuscript fully available?

Reviewer #1: Yes

Reviewer #2: Yes

Reviewer #3: Yes

Reviewer #4: Yes

4. Is the manuscript presented in an intelligible fashion and written in standard English?

Reviewer #1: Yes

Reviewer #2: Yes

Reviewer #3: Yes

Reviewer #4: Yes

5. Review Comments to the Author

Reviewer #1: This is a clearcut paper, with logical structre and meaningful results.

As a small discrepancy, the sentence at lines 169-170 needs revision, perhaps the authors mean " cohort of men with identified hyperprolactinemia and

REDUCED testosterone levels. "

Reviewer #2: Overall, the manuscript is good.

*Please add the date of the ethical approval.

*Please rephrase the lines 169-170 for clarity.

*Please provide if there are higher resolution images for radiologic evaluation.

Reviewer #3: This manuscript titled "Idiopathic hyperprolactinemia-associated hypogonadism in men presenting with normal

testosterone levels" is a new addition to the literature. It highlights recognition of unique clinical cases. However, the small number of cases (23) may create potential need for a larger sample size to confirm your findings.

Still this manuscript fills the gap in scientific knowledge.

Reviewer #4: The work is commendable for highlighting a rare clinical presentation, for providing detailed clinical, hormonal, and imaging data, and for suggesting pituitary height as a potential diagnostic and prognostic marker.

However, I believe the following issues should be addressed to strengthen the manuscript:

1. Interpretation of Results: The conclusion regarding cabergoline’s superiority should be moderated, as the observed differences were not statistically significant.

2. General grammar corrections and formatting improvements are needed to enhance the manuscript’s clarity and consistency.

• Line 41: “…but whose clinical profiles were less well-defined.”

→ Consider removing “less” → “…but whose clinical profiles were poorly defined.”

• Table 1: “Weekness” appears multiple times → but the correct spelling should be “Weakness”

• Units: “ng/ml” is sometimes written inconsistently (e.g., “ng/ml” vs “ng/mL”). Standardize.

• Numbers: inconsistent use of “±” formatting (e.g., “6.400 ± 0.9653 mm” vs. “6.4 ± 0.97 mm I suggest rounding to 2 decimal places to make them consistent.

• Abbreviations: Ensure all abbreviations (e.g., BRC, CAB, ED, IIEF-15) are defined at first use and used consistently.

3. Discussion: The suggestion that “normal” testosterone levels may still represent relative insufficiency for physiological needs warrants clearer integration into the discussion.

4. Limitations: The retrospective design, small sample size, and reliance on a subjective questionnaire (IIEF-15) should be emphasised more prominently as constraints on interpretation and generalizability.

5. Future Directions: Consider recommending prospective or multicentre studies to validate pituitary height as a biomarker and further evaluate treatment outcomes.

Overall, the manuscript is of interest and, following minor revisions, could make a valuable contribution to the literature on hyperprolactinemia and male hypogonadism.

6. PLOS authors have the option to publish the peer review history of their article (what does this mean? ). If published, this will include your full peer review and any attached files.

**Do you want your identity to be public for this peer review?** For information about this choice, including consent withdrawal, please see our Privacy Policy .

Reviewer #1: No

Reviewer #2: No

Reviewer #3: **Yes: ** Ayman S. Amer

Reviewer #4: No

---

## [Author Response · Author response to Decision Letter 1]

4 Sep 2025

Dear Dr. Bustani and Reviewers,

We sincerely thank you and the reviewers for your thorough evaluation of our manuscript (Manuscript No.: PONE-D-25-39935) entitled “Idiopathic hyperprolactinemia-associated hypogonadism in men presenting with normal testosterone levels”. We appreciate the positive feedback and valuable suggestions, which have helped us improve the clarity and quality of our work. In accordance with the Academic Editor’s and Reviewers’ comments, we have revised the manuscript as detailed below. All changes are indicated in the marked-up manuscript (with tracked changes), and line numbers refer to the revised clean manuscript for ease of reference. We have also ensured that the manuscript now fully adheres to PLOS ONE formatting and policy requirements, including data availability and funding disclosure.

Below we address each point raised by the Academic Editor and four reviewers point-by-point, with our responses in bold and references to specific changes in the manuscript.

Academic Editor: “Your manuscript is of interest and well-structured. Please revise by clarifying the sentence at lines 169–170, moderating conclusions on cabergoline (as differences were not statistically significant), and improving grammar, formatting, and consistency (spelling, units, abbreviations). Expand on the limitations (retrospective design, small sample, use of IIEF-15) and suggest future directions. Ensure the ethics approval date is included, and provide higher resolution images if available.”

Response: We are grateful for the editor’s encouraging remarks and guidance. We have carefully addressed all the points noted:

Clarification of lines 169–170: We have rewritten the sentence in the Results (Patient Characteristics) to clearly convey that the 23 patients in our study represent approximately 10% of all male hyperprolactinemia cases seen in that period, with the majority of hyperprolactinemic men exhibiting reduced testosterone levels. This correction resolves the ambiguity (see lines 168–170 in the revised manuscript).

Moderation of conclusions about cabergoline: We have tempered our language wherever cabergoline’s efficacy is discussed. In the Discussion and Conclusion, we no longer state that cabergoline is unequivocally “more effective” or “preferred” over bromocriptine. Instead, we acknowledge that while cabergoline showed a trend toward faster prolactin normalization, our data did not show statistically significant differences in overall outcomes between cabergoline and bromocriptine. For example, we revised the Conclusion to remove the phrase “cabergoline emerging as the preferred therapeutic agent” (now on lines 345–347). We believe this more cautious wording accurately reflects the non-significant differences and addresses this concern.

Grammar, spelling, and formatting: We performed a thorough edit for language and consistency. Notably, we corrected the typo “weekness” to “weakness” in Table 1, standardized units (e.g., consistently using “ng/mL” instead of mixed “ng/ml” or “ng/mL”), and ensured numerical data are rounded to two decimal places throughout (for instance, pituitary height is now reported as 6.40 ± 0.97 mm on line 180, instead of 6.400 ± 0.9653, for consistency). We also checked that all abbreviations are defined at first use and used uniformly (e.g., ED, IIEF-15, PRL, BRC, CAB, etc.). These edits improve the manuscript’s clarity and conformity to journal style.

Expanded limitations: We have enhanced the Limitations section of the Discussion (lines 324–333) to more prominently acknowledge the study’s retrospective design, the small sample size (n=23), and the reliance on a subjective questionnaire (IIEF-15). We explicitly note that these factors limit the generalizability of our findings and that the inherent subjectivity of self-reported sexual function is a constraint. This expansion underscores the need for caution in interpretation.

Future directions: In line with the suggestion, we added a statement about future research at the end of the limitations paragraph (lines 333–337). We now recommend prospective and multi-center studies to validate the significance of pituitary height as a biomarker and to confirm our findings in larger populations. This provides a forward-looking perspective and addresses the reviewer’s point about the importance of further validation.

Ethics approval date: We have included the date of ethics committee approval in the Methods section. The manuscript now reads that the study was approved by the Ethics Committee of People’s Hospital of Gaozhou on 25/07/2024 (Approval No. GYLLPJ-2024080) (see line 110).

Image resolution: Due to the compression of the PDF file generated in the submission system, the MRI images are unclear. The original images we uploaded in the submission system are high-resolution. All figure images have been checked with PLOS’s PACE tool to ensure they meet the journal’s quality requirements. (Figures in the revised submission are high resolution, though this is not reflected in the text of the manuscript.)

We trust that these revisions fully address the Academic Editor’s concerns. Detailed responses to each reviewer comment are provided below.

Reviewer #1: “This is a clear-cut paper, with logical structure and meaningful results. As a small discrepancy, the sentence at lines 169–170 needs revision, perhaps the authors mean ‘cohort of men with identified hyperprolactinemia and reduced testosterone levels.’”

Response to Reviewer #1: We thank Reviewer #1 for the positive feedback on our manuscript’s clarity and significance. We have addressed the noted discrepancy:

Clarification of lines 169–170: We agree that the original phrasing was confusing. The sentence in question (Results section, patient cohort description) has been revised for clarity. It now reads: “The 23 patients in our study represent approximately 10% of all men diagnosed with hyperprolactinemia at our center during the study period, with the remaining ~90% presenting with the more typical finding of reduced testosterone levels.” (lines 168–170). This change makes it explicit that most men with hyperprolactinemia have low testosterone, and that our 23 cases were the minority who had normal testosterone despite hyperprolactinemia. We believe this addresses the reviewer’s suggestion and clarifies the intended meaning.

We appreciate the reviewer bringing this to our attention, and we hope the revision resolves the discrepancy.

Reviewer #2: “Overall, the manuscript is good. (1) Please add the date of the ethical approval. (2) Please rephrase the lines 169–170 for clarity. (3) Please provide if there are higher resolution images for radiologic evaluation.”

Response to Reviewer #2: Thank you for your overall positive assessment and these specific suggestions. We have implemented all the requested changes:

Ethics approval date: We have added the date of ethical approval to the Methods. It now states that the study was approved by the ethics committee on 25/07/2024 (Approval No. GYLLPJ-2024080) (see line 110). Including the approval date provides full transparency of our ethical clearance, as requested.

Rephrasing lines 169–170: We have rewritten the sentence in question to improve clarity (see lines 168–170, as also noted in our response to Reviewer #1). The new wording clearly conveys that the 23 patients with idiopathic hyperprolactinemia and normal testosterone constituted ~10% of all male hyperprolactinemia cases in that period, and that the majority of cases had reduced testosterone. This rephrasing eliminates the ambiguity in the original phrasing.

Higher resolution images: Due to the compression of the PDF file generated in the submission system, the MRI images are unclear. The original images we uploaded in the submission system are high-resolution. All figure images have been checked with PLOS’s PACE tool to ensure they meet the journal’s standards. We trust that the image quality is now satisfactory for radiologic evaluation.

We appreciate the reviewer’s suggestions and believe these updates have improved the manuscript’s completeness and clarity.

Reviewer #3: “This manuscript is a new addition to the literature. It highlights recognition of unique clinical cases. However, the small number of cases (23) may create potential need for a larger sample size to confirm your findings. Still, this manuscript fills the gap in scientific knowledge.”

Response to Reviewer #3: We thank the reviewer for recognizing the novelty and importance of our findings. We agree that the small sample size is a limitation and have taken steps to underscore this in the revised manuscript:

We have emphasized the small sample size and its implications in the Discussion’s limitations section. Specifically, we note that our cohort of 23 patients is modest and that this limits statistical power and generalizability (see lines 324–333). We stress that larger studies are needed to confirm the findings. This directly addresses the reviewer’s point about the need for a larger sample to validate our observations.

We have also added a forward-looking statement that future prospective or multicenter studies should be conducted (lines 333–337), acknowledging that confirmation in a broader population would strengthen the evidence.

We appreciate the reviewer’s supportive remarks that our study fills a knowledge gap. By highlighting the limitations and recommending future research, we aim to provide a balanced interpretation of our results. Thank you once again for your encouraging evaluation.

Reviewer #4: *“The work is commendable for highlighting a rare clinical presentation, providing detailed data, and suggesting pituitary height as a marker. However, I believe the following issues should be addressed to strengthen the manuscript:

1.Interpretation of Results: The conclusion regarding cabergoline’s superiority should be moderated, as the observed differences were not statistically significant.

2.General grammar/formatting: Improvements needed for clarity and consistency.

Line 41: change “less well-defined” to “poorly defined.”

Table 1: “Weekness” → “Weakness.”

Units: standardize “ng/ml” vs “ng/mL.”

Numbers: format consistently (e.g., use 2 decimal places throughout).

Abbreviations: Ensure all (BRC, CAB, ED, IIEF-15, etc.) are defined at first use and used consistently.

3.Discussion: Clarify that “normal” testosterone levels may still represent relative insufficiency for physiological needs.

4.Limitations: Emphasize retrospective design, small sample size, and reliance on subjective questionnaire (IIEF-15) as limitations affecting interpretation and generalizability.

5.Future Directions: Suggest prospective or multicenter studies to validate pituitary height as a biomarker and further evaluate outcomes.”*

Response to Reviewer #4: We are grateful for the reviewer’s thorough and insightful comments. We have implemented all suggested changes, as detailed below:

1.Moderating cabergoline conclusions: We completely agree and have tempered our interpretation regarding cabergoline’s efficacy. In the Discussion (Results interpretation, line 235) we no longer claim cabergoline is unequivocally “more effective.” The relevant sentence now states that cabergoline showed a trend toward greater efficacy (e.g., shorter treatment duration) but that these differences were not statistically significant in our cohort (see lines 315–319). Likewise, in the Conclusion (lines 345–347), we removed the statement that cabergoline is the “preferred therapeutic agent.” Instead, we note that both cabergoline and bromocriptine were effective in restoring function, and we explicitly state that no significant difference was observed in overall outcomes between the two. These changes ensure our conclusions are appropriately cautious and supported by the data.

2.Grammar, spelling, and formatting fixes: We conducted a careful review of the manuscript and addressed all the points raised:

Abstract, line 41: We changed “less well-defined” to “poorly defined” (see line 41).

Table 1: The typo “Weekness” has been corrected to “Weakness” in every instance in the table (e.g., Patient 9’s initial complaint now reads “Low libido, Weakness”).

Units: We standardized units to a consistent format. All occurrences of “ng/ml” have been changed to “ng/mL” (e.g., line 48, 173, 174), and we ensured consistent formatting of other units (mm, years, etc.) according to journal style.

Numerical formatting: We made formatting of all numeric data consistent. Means and standard deviations are now presented to two decimal places throughout (unless an integer value). For example, pituitary height is given as 6.40 ± 0.97 mm (line 180) instead of 6.400 ± 0.9653, and treatment durations are rounded (e.g., cabergoline treatment 2.40 ± 1.14 months, line 221). We also added missing spaces and corrected any formatting issues with the “±” symbol to ensure uniform appearance.

Abbreviations: We verified that all abbreviations are defined at first use and used consistently thereafter. For instance, we introduce cabergoline (CAB) and bromocriptine (BRC) in the Introduction (lines 77–78), erectile dysfunction (ED) in the Introduction (line 73), and International Index of Erectile Function-15 (IIEF-15) in the Methods (line 105). Throughout the text and tables, we now use the abbreviations uniformly (e.g., using “CAB” and “BRC” rather than switching back to the full drug names arbitrarily). We also ensured consistency in terms like “hyperprolactinemia-associated hypogonadism” and removed any redundant abbreviations. These corrections collectively improve the manuscript’s readability and professional presentation.

3.Discussion – “normal” testosterone levels and physiological sufficiency: We have expanded the discussion to explicitly address this point. In the revised Discussion (lines 274–277), we added an explanation that normal-range testosterone levels in these patients may still have been insufficient for their physiological needs. We state that the baseline “normal” testosterone was relatively inadequate for normal sexual function, as evidenced by the improvement in symptoms and upward trend in testosterone after treatment. The text now reads: “This suggests that the ‘normal’ testosterone levels observed initially were not truly sufficient for these patients’ androgen requirements (i.e., a relative testosterone insufficiency), and that normalizing prolactin relieved its inhibitory effect on the hypothalamic–pituitary–gonadal axis.” (lines 274–277). This addition integrates the concept of relative androgen insufficiency despite normal total testosterone, directly addressing the reviewer’s concern.

4.Limitations: We have reinforced the limitations section to highlight the retrospective design, small sample size, and use of a subjective questionnaire. The revised paragraph (lines 324–333) now explicitly mentions each of these factors: We note that the IIEF-15, while validated, is subjective and may introduce bias; we underscore that the retrospective nature and the modest sample size (n=23) limit our ability to draw broad conclusions; and we caution that these limitations impact the interpretation and generalizability of our findings. By spelling out these issues, we align the discussion with the reviewer’s suggestions and ensure readers are fully aware of the study’s constraints.

5.Future directions: In accordance with the reviewer’s advice, we added a forward-looking statement about future studies (see lines 333–337). We suggest that prospective, ideally multicenter, studies be conducted to further investigate this phenomenon. In particular, we mention the need to validate pituitary height as a biomarker for diagnosing and monitoring idiopathic hyperprolactinemia-associated hypogonadism. This addition not only addresses the reviewer’s point but also provides a logical conclusion to our discussion by indicating how the research can be expanded upon.

Overall, we believe these revisions have significantly strengthened the manuscript. We are grateful to Reviewer #4 for the careful critique. The manuscript is now more precise in its language and more comprehensive in its discussion of limitations and implications.

Additional Revisions (Journal Requirements)

---

## [Editor Report · Decision Letter 1]

8 Sep 2025

Idiopathic hyperprolactinemia-associated hypogonadism in men presenting with normal testosterone levels

PONE-D-25-39935R1

Dear Dr. Cheng,

We’re pleased to inform you that your manuscript has been judged scientifically suitable for publication and will be formally accepted for publication once it meets all outstanding technical requirements.

Kind regards,

Ghadeer Sabah Bustani, Ph.D

Academic Editor

PLOS ONE
---

## [Editor Report · Acceptance letter]

PONE-D-25-39935R1

PLOS ONE

Dear Dr. Cheng,

I'm pleased to inform you that your manuscript has been deemed suitable for publication in PLOS ONE. Congratulations! Your manuscript is now being handed over to our production team.

Kind regards,

on behalf of

Dr. Ghadeer Sabah Bustani

Academic Editor

PLOS ONE